# Cross-sector user and provider perceptions on experiences of shared-care clozapine: a qualitative study

Camilla Sowerby, Denise Taylor

Department of Pharmacy and Pharmacology, University of Bath, Bath, UK

**Correspondence to**
Mrs. Camilla Sowerby;
camilla.sowerby@berkshire.nhs.uk

## ABSTRACT

**Objectives** (1) To explore individual perceptions on experiences of people receiving and/or delivering a shared-care clozapine service and (2) to gain an understanding of effectiveness and acceptability of shared-care clozapine.

**Design** Interpretative phenomenological analysis guided the delivery and analysis of a semistructured interview and focus group study designed to explore participant experience of shared-care clozapine. Ethical approval 13/EM/0286 was gained in July 2013 from East Midlands—Nottingham 1 REC.

**Participants** Eight stakeholder groups from Adult and Forensic Mental Health involved in shared-care clozapine provision delivered in primary care were identified for recruitment from one mental health trust in England (six different groups of healthcare professionals (HCPs), clozapine service users (CSUs) and their carers). To be eligible for recruitment, all potential participants had to be either providing, receiving or the carer of a person receiving clozapine by shared care.

**Results** 32 HCPs and 6 CSUs were recruited and 14 interviews and 6 participant homogenous focus groups were run. Four shared superordinate themes were identified: Clozapine Process, The Sharing of Care, The Provision of Care and Multi-professional Relationships. Differences between Adult and Forensic engagement in shared care were noted and both HCP and CSU relationships were mapped to the Wish conceptual framework of relationships to provide insight into how shared-care clozapine can provide a mechanism for provision of person-centred care, which was present in the Forensic HCP–CSU but not General Adult HCP–CSU relationship.

**Conclusions** The Forensic HCP/CSU relationship demonstrated how cross-sector working through shared-care clozapine can provide a mechanism for provision of person-centred care by enabling a person-centred focus to care delivery which supported CSUs to live as independently as possible. Person-centred care demonstrably improves patient care outcomes and wider implementation of shared-care clozapine could provide greater integration of people with serious mental illness and reduce stigma within the community while improving patient outcomes.

## INTRODUCTION

In recent years, a number of UK government documents have been published concerning the provision of healthcare and mental health services, including: Equity and Excellence: liberating the National Health

### Strengths and limitations of this study

► This is the first qualitative study to explore the experiences and perceptions of those who use and provide a shared-care clozapine service.
► Identification of potential participants relied on two information sources being up to date: the Denzapine Monitoring Service website and the electronic medical notes.
► In terms of group size, clozapine service users (CSUs) were not equally represented in number; there were more Forensic community mental health team (CMHT) CSUs compared with General Adult CMHT CSUs. This may mean the experiences and perceptions of CSUs from General Adult CMHT may not be sufficiently represented. The CSU recruitment method required them to be invited to the research by their own clinical team. The greater recruitment of CSUs from the Forensic CMHT could be related to their enabling approach to care or a reflection of the team's experience of shared care in comparison with General Adult CMHT.
► No carers of CSUs participated, reflecting the isolation from family and loved ones that some people with serious mental illness live with.
► Data validity is reflected in the ability to substantiate the analysis with participant quotes and provide a dialogue of discussion between the results and existing literature.

Service (NHS),[1] No Health without Mental Health[2] and the Five Year Forward View.[3] The main focus of these publications is there should be choice in how patients obtain their care, and care should be individualised and recovery focused to improve patients' independence.

Clozapine is the only antipsychotic with efficacy in treatment-resistant schizophrenia and is effective in around one-third of these patients.[4 5] Additionally in the UK, due to the risk of clozapine-associated agranulocytosis, there is a requirement to undertake regular full blood count (FBC) monitoring and associated limitations on amount of medication supplied.[6] Additionally. in the UK. all prescribers, organisations dispensing

**Table 1** Description of models used to provide clozapine

| | Community mental health team (CMHT)-based supply | Clozapine clinic | Shared care |
|---|---|---|---|
| Full blood count (FBC) monitoring | The FBC monitoring is undertaken by general practitioners (GPs) and either sent to the local pathology laboratory or posted to the clozapine company for processing. The results are automatically uploaded onto the Denzapine Monitoring Service website if sent to them for processing, while hospital pharmacy upload those results of FBCs sent to the local pathology laboratory. | Clozapine patients attend the clinic, which is held at a hospital site, where FBC monitoring is undertaken and sent to the clozapine company for processing. | The FBC monitoring is undertaken by GPs and either sent to the local pathology laboratory or posted to the clozapine company for processing. |
| Prescription | The prescriptions are managed by hospital pharmacy and written by the responsible clinician (RC) of the clozapine patient. The patient maintains regular outpatient appointments with his/her RC. The frequencies of these depend on the individual patient. | Same as CMHT-based supply. | The GP prescribed clozapine on an FP10 prescription. The clozapine service user (CSU) maintains regular outpatient appointments with his/her RC. The frequencies of these depend on the CSU. CSUs can only use the shared-care service if they have been on clozapine for 12 months and are stabilised on treatment. |
| Dispensing | Hospital pharmacy dispenses clozapine to the relevant CMHT site or posts the medicine to the patient. Patients either collect their prescription from the CMHT site or CMHT staff to deliver to them. | Hospital pharmacy dispenses clozapine to the clozapine clinic. Patients collect their clozapine from the clinic. | Clozapine is supplied against the FP10 by the community pharmacy the CSU wishes to use. The community pharmacy needs to be registered with the clozapine company in order to be able to supply clozapine. CSUs either collect their prescription from the community pharmacy or use their delivery services. |

clozapine and people taking clozapine have to be registered with a brand-specific clozapine website dependent on trust contract. An alternative model of clozapine provision to the usual clozapine clinic model or community mental health team (CMHT)-based supply is shared-care clozapine (see table 1).

## METHODOLOGY AND METHODS
In this model, clozapine service users (CSUs) obtain their FBC monitoring, prescription and clozapine supply from their general practitioner (GP) and community pharmacy, respectively. In shared care, the responsibilities for clozapine provision are shared between specialist mental health and primary care healthcare professionals (HCPs). Evidence suggests that shared-care clozapine is effective in maintaining the monitoring and management of clozapine in a less restrictive care pathway.[7] Studies have highlighted that shared-care clozapine is not suitable for all CSUs and propose criteria to identify those most suitable.[7 8]

A growing body of evidence of the benefits of a shared-care approach in other long-term conditions, including serious mental illness, is evolving.[9 10] Currently, no published literature explores the perceptions of experiences of those who use and/or provide a shared-care clozapine service. The importance of patient experience of healthcare services is increasingly recognised in healthcare policy, to shape and inform service development, and payment structures for providers. Through understanding individuals' perceptions and experiences of shared-care clozapine, insight can be gained into whether shared-care clozapine fulfils the government's vision for mental healthcare and inform future service developments to improve quality and patient experience.

### Aims of the study
1. To explore individual perceptions on experiences of people receiving and/or delivering a shared-care clozapine service.
2. To gain an understanding of effectiveness and acceptability of shared-care clozapine.

### Ethical approval
This study took place within one NHS mental health trust in the UK and a favourable NHS ethics committee opinion (East Midlands—Nottingham 1 NRES: Reference 13/

**Table 2** Focus group and interview topic guides

| Clozapine service user topic guide | | Healthcare professional (HCP) topic guide | |
|---|---|---|---|
| **Questions** | **Probing questions** | **Questions** | **Probing questions** |
| When you were asked about participating in the shared-care clozapine service, what were your first thoughts? | Why did you agree to participate in the clozapine shared-care service? How did it make you feel? What does the clozapine shared-care service mean to you? | What do you think the differences are in care for someone if they receive their clozapine through the shared-care clozapine service compared with other means, for example, secondary care? | What do you think are the benefits and negatives? What do you think was trying to be achieved with the shared-care agreement? What do you think is actually achieved with the shared-care service? What does this mean for you? |
| What is your experience of being involved in the shared-care clozapine service? | Describe the differences to you by obtaining your clozapine through the shared-care service. Can you identify positive and negative influences on your experience? How does that make you feel? What does that mean to you? How has the shared-care clozapine service impacted you? How have your thoughts and feelings towards the clozapine shared-care service changed over time? | What is your experience of being involved in the shared-care clozapine service? | What were your first thoughts and feelings when you were asked to participate? Can you identify positive and negative influences on your experience? How does that make you feel? What does that mean to you? How has the shared-care clozapine service impacted on you? |
| Describe the roles of the HCPs who provide the clozapine shared-care service. | How has the clozapine shared-care service affected your thoughts and feelings about the HCPs providing the clozapine shared-care service? Has the shared-care service changed your relationship with these professionals? In what way has the clozapine shared-care service affected thoughts and feelings about these HCPs? | Can you describe your role within the shared-care clozapine service? | What does your input mean to you? How does it make you feel? How has the shared-care clozapine service affected your working relationship with other professionals? What does this mean for you? How has the shared-care clozapine service developed you as a professional? What training/support have you received? How did you receive this training? What other ways would you like to get more involved in the care of these patients? |

EM/0286), and local research and development approval was received in 2013.

### Study design

Interpretative phenomenological analysis (IPA) was selected as the method as it attempts to explore the personal experience as it is perceived by the participant, rather than the researcher producing an objective statement of the event itself. IPA also allows for the expertise of the researcher to be acknowledged by reflexivity in the analytical processes.[11] Semistructured interviews and focus groups methods were used to explore perceptions and experiences of CSUs, their carers, GPs, community psychiatric nurses (CPNs), social workers (SWs),

community and hospital pharmacy staff and responsible clinicians (RCs) (see table 2 for topic guide).

The HCP topic guide was piloted with pharmacist focus group participants and amended following feedback. Topic guides were also reviewed after each interview or focus group if questions were found to be too direct, leading or closed. To be eligible for recruitment, all potential participants had to be either providing, receiving or the carer of a person receiving clozapine by shared care. Carers of CSUs were only eligible for recruitment if the CSU they cared for invited them to participate in the research.

Eligible CSUs, RCs and community pharmacists were identified via the Denzapine Monitoring Service (DMS)

website. GPs were identified from CSUs electronic notes by the researcher Camilla Sowerby (CS) and hospital pharmacist participants were identified by the dispensary manager. All General Adult and Forensic CMHTs were targeted by the researcher to potentially recruit CPNs and SWs working in shared-care services, and potential CSU participants were approached by their care team. All potential participants received an information pack about the study, including an expression of interest form. and given a minimum of 2 weeks to make their decision.

All interviews and focus groups were to be completed by CS and held at a convenient time and place for participants, either in a room at their place of work or on a mental health trust site, where participants could speak confidentially. Only researchers and participants were to be present during data collection and participant consent was to be received immediately prior to them taking part. Each participant received a gift voucher as a token of appreciation for their time and input into the study. Interviews and focus groups were audio-recorded and transcribed verbatim to ensure a faithful reproduction of the participants' participation and anonymised by removing identifiers. Transcripts were assigned an descriptor, for example, Doctor 2, to enable comparison of experiential perspectives to be made during analysis. In the findings, these were changed to pseudonyms to protect confidentiality.

### Researchers' personal characteristics

During this study, CS (MPharmS) was a clinical mental health pharmacist in forensic services in the mental health trust where the study was conducted and the study formed part of her postgraduate MSc research project. She completed relevant theoretical and experiential training in qualitative research methods, particularly in IPA. CS completed all interviews and focus groups and was observed on two occasions by Denise Taylor (DT) for appropriateness and effectiveness.

DT (PhD, MSc, MPharmS, FFRPS, MCMHP) was a senior lecturer and research supervisor for CS, providing training in qualitative research methods, analyses, informed consent and capacity assessment.

### Relationship with participants

Due to her role as clinical pharmacist, CS had developed professional relationships with some (but not all) of the primary and secondary care HCPs prior to the study; she did not know any of the CSUs. All participants were aware this study was part of a further degree. IPA enables researcher expertise to be acknowledged through reflexivity while simultaneously preserving the voice of the participant.

DT knew none of the study participants but had previously worked in and researched delivery of clozapine services. Reflexivity throughout the study enabled personal and professional bias to be recognised and addressed.

### Data analysis

As this research explored individuals' experiences and perceptions, IPA was used. The intention of IPA is to explore the participants' experience as perceived by them and reflect this in the interpretation by the researcher.[12] In line with accepted analytical process for IPA, each transcript was manually coded line by line into themes by CS. Robustness of the analytical process was assessed by the coding of a transcript by both CS and DT. There was a high level of agreement regarding coding and CS continued with the analytical process. At the end of the coding process, CS and DT met to discuss and clarify emerging themes. Once superordinate or 'master' themes were identified for each stakeholder participant group, the analysis moved to looking at connections and patterns across all participant groups. No software was used to aid the analysis of data or themes.

## RESULTS

A total of 38 participants were recruited, including 32 HCPs and 6 CSUs (General Adult CMHT=1; Forensic=5), but no carers as no expressions of interest to participate were received. It is not known how many potential participants were approached by members of the CMHT and decided not to participate in the study, but once recruited to the study there were no dropouts. Fourteen interviews and six homogenous focus groups from General Adult and Forensic CMHTs were completed. Demographics of participants can be found in table 3.

The duration of interviews and focus groups ranged from 30 to 90 min and none were repeated. Data saturation was achieved as no new themes emerged during the analysis of final transcripts and field notes were taken for each interview or focus group to support reflexivity. Due to limitations in time and funding, transcripts and findings were not returned to participants for comment. The findings below describe participant perceptions of their experiences of a shared-care clozapine service.

Four superordinate themes emerged from the data: The Sharing of Care, Multi-professional Relationships, The Provision of Care and the Clozapine Process. The interconnectedness of these themes and minor themes is illustrated in figure 1.

Excerpts from participant transcripts (shown in indented italic text) have been chosen to exemplify experiential perceptions and changed as little as possible to retain data authenticity. If additions were made for clarification, these are bracketed within the quote and non-italicised. When the term Forensic HCP or General Adult HCP is used, Forensic and General Adult CMHT HCP is being referred to respectively.

### Clozapine Process

Clozapine is a unique antipsychotic with regard to its prescribing, monitoring and supply; therefore,

| Table 3 | Demographics of participants | | |
| --- | --- | --- | --- |
| Participants* per focus group or interview | Profession | Sector | Years with shared-care clozapine |
| Mark | Hospital pharmacist | Hospital pharmacy, secondary care | 1.5 |
| Emily | MMT† | | 2 |
| Laura | MMT† | | 5 |
| Angela | Receptionist | | 5 |
| Luke | Community psychiatric nurses (CPNs) | Forensic community mental health team (CMHT), secondary care | 2 |
| Bethany | | | 2 |
| John | | | 2 |
| Kate | CPNs | General Adult CMHT, secondary care | 4 |
| Anne | | | 4 |
| Denise | | | 4 |
| Jennifer | CPN , social worker (SW) | General Adult CMHT, secondary care | 2 |
| Trace | CPNs | | 5 |
| Victoria | | | 1.5 |
| Lucy | All SWs | Forensic CMHT, secondary care | 2 |
| Jane | | | 2 |
| Nick | | | 2 |
| Claire | | | 2 |
| Tim | All clozapine service users (CSUs) | Attached to the Forensic CMHT | 1–2 |
| Simon | | | 1–2 |
| Adam | | | 1–2 |
| Frank | | | 1–2 |
| Rob | | | 1–2 |
| Richard | CSU | Attached to General Adult CMHT | Unknown |
| Dr Harrison | Responsible clinician (RC) | Forensic | 2 |
| Dr Brown | RC | Forensic | 2 |
| Dr Taylor | RC | Forensic | 2 |
| Dr Smith | RC | General Adult | 4 |
| Charles | Community pharmacist | Primary care | Unknown |
| Catherine | Community pharmacist | Primary care | 2–3 |
| Margaret | Community pharmacist | Primary care | 4 |
| Tom | Community pharmacist | Primary care | 9 months |
| Sophie | Community pharmacist | Primary care | Unknown |
| Marie | Community pharmacist | Primary care | 4–5 |
| Dr Hudson | General practitioner (GP) | Primary care | 2 |
| Dr White | GP | Primary care | 10 |
| Dr Green | GP | Primary care | 2 |

*Pseudonym names used for confidentiality.
†Medicines management technician.

knowledge of all the components of prescribing clozapine and its possible consequences are important factors in a clozapine supply service.

## Knowledge
The process involved in prescribing and supplying clozapine is different to any other antipsychotic; therefore, it takes time and experience to understand and become familiar with its nuances. Greater participant knowledge of the process meant that HCPs understood why things might go wrong, enabling them to have greater confidence in resolving these issues and their role within that.

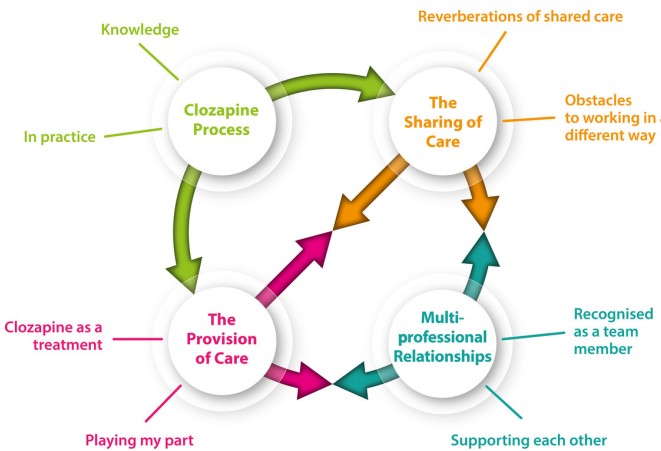

**Figure 1** Illustration of superordinate themes and their connections.

"I looked up some information and used the BNF and it was I guess somewhat hesitant, clearly it is medication that has got some significant potential side effects and risks with it. But umm I think that I felt like I have got a good working relationship with the mental health team and working with the patient that I have on clozapine, and so was able to talk with the (hospital) pharmacist and address any issues." Dr Green, GP

"I suppose it's enlarged my experience so I'm more confident with… …once you do more than one customer your confidence increases and you find out the….and you can deal with customers easier can't you? So professionally that way and also you're you feel confident to deal with those patients and other professionals as well" Marie, community pharmacist

### In practice
Initially, HCPs perceived the CSU required certain skills such as organising and planning for shared-care clozapine to be successful, so they used surrogate markers of stability and capability to identify suitability of CSUs. With experience of working with shared care, Forensic HCPs realised that CSUs did not need to be already stable or capable in order for shared care to be successful. They found CSUs responded well to the challenge of being responsible for obtaining their own FBC, repeat prescription and supply of clozapine.

"A (CSU) as well who was really chaotic initially, and we were really worried about whether he'd manage it all, but it's actually made him get a lot more organised."Bethany, Forensic CPN

HCPs and CSUs perceived the time frame between obtaining a valid FBC and supplying clozapine before the last supply ran out as being very tight. This became a source of anxiety for both HCPs and CSUs, as they were extremely aware of the potential negative consequences of readmission to hospital for retitration with a delayed supply of clozapine. Circumstances which could delay the supply of clozapine were perceived to occur relatively frequently by community pharmacists.

"….I guess it's just because there's such a tight schedule of the drug, it gives you no leeway for these kind of exceptional circumstances which do happen reasonably regularly." Charles, community pharmacist

Limitations to GP appointment systems were identified, as for some it was not possible to book an appointment 4 weeks in advance for a FBC. This meant CSUs needed to adapt their routine to ensure they remembered to book their appointment at a later date and when sufficient appointments were still available.

"They give it up to like…I think it's 2 weeks maximum but they won't do a full 28 days, but like (I) write down on a calendar today every month when my calendar starts, need to book the blood test and normally for the week beforehand to book it." Adam, CSU

### The Sharing of Care
The sharing of care reverberated far wider than just the prescribing and monitoring of clozapine and was associated with its own obstacles as it required individuals to work in a different way.

#### Reverberations of shared care
Some HCPs thought that the drive for shared-care clozapine was one of cost saving; however with experience, Forensic HCPs identified greater CSU independence as their driver for engagement. For Forensic HCPs, shared care enabled CSUs to continue to develop skills required for independence and a new opportunity for them to demonstrate trust in the CSU. This trust led to a change in the relationship dynamic and activated the CSU to take responsibility for their care and engage in the process.

"Initially it was savings because…quite some time once I saw how the first person was working and did….once I saw the difference it made with him, how then we felt about him taking medication and how he was responding to that, I think then I moved away from seeing it as a cost saving exercise and seeing it as something that could actually useful in maintaining independence for the client." Nick, Forensic SW

This was supported by CSUs views that shared care was a supportive opportunity to develop their independence and take back ownership of their health, which they felt in turn boosted their self-esteem. CSUs also perceived that engaging in shared care was normalising and reduced stigma associated with mental illness and clozapine treatment.

"The emphasis is all on you sort of thin in the community, whereas before it was thought that people with mental illness couldn't live in the community properly without having problems and things like

that, whereas people are seeing especially the people on Clozapine, that they can live a good normal active life in the community…" Adam, CSU

The physical health of CSUs was valued as equally important to their mental health and shared care provided GP involvement in clozapine prescribing with awareness of the need to manage negative effects of clozapine on physical health. Both Forensic HCPs and CSUs perceived the GP as the expert in physical health and appreciated and benefitted from this more holistic approach to care. In contrast, GPs not involved in shared care may not even be aware their patient receives clozapine from a mental health trust and any possible associated negative iatrogenic effects. At the time of this study, shared-care records between primary and secondary care were not widely used.

"You can't separate off the, you know in the case of these patients, the mental health issues from the physical health and their overall wellbeing, emotional spiritual wellbeing, it is all their whole entity part of the whole and interplay with one another." Dr Green, GP

### Obstacles to working in a different way
Shared-care clozapine required secondary care HCPs to trust primary care HCPs they may not have previously met and for primary care HCPs to take on the responsibility of an unfamiliar medication with very restrictive prescribing and supply requirements. As mentioned in the In practice section, there was initial apprehension on possible negative consequences of a missed clozapine dose. Apprehensions from primary care HCPs eased over time due to growing relationships with secondary care HCPs providing a sense of support and a true sharing of care.

"It's not a drug I'm totally at ease with, so there are apprehensions when any specialist says 'please do this, these are the protocols' and unless you really feel you've got a backup, you feel somewhat exposed. That hasn't particularly been an issue because the support has been there." Dr White, GP

Often General Adult HCPs described negative assumptions on the abilities and/or expectations of CSUs in relation to independence and involvement in managing their own medication. This affected their ability to trust CSUs in taking on this responsibility or even seeing a need for shared care.

"I suppose (it) depends how well they are doesn't it, how insightful they are in the first place, how sassy they are with how systems work and stuff really. At the end of the day, they just want their meds to turn up when they need more. (The CPN) comes round at 4 with the meds, that's what they want isn't it?" Jane, General Adult CPN

### The Provision of Care
Although clozapine is the active treatment, the care provision included the role each member of the multi-professional team played in supporting the recovery and management of the individual.

### Clozapine as a treatment
Many HCPs and all CSUs thought that clozapine often provided the best opportunity for recovery and hospital discharge in individuals with schizophrenia; particularly if they had already tried other antipsychotics without success. These highly valued benefits of clozapine provided the motivation for CSUs to continue taking it despite its negative effects.

"The other thing with Clozapine which was explained to me is side effects and pluses and minuses and sort of thing, but it's like to me it's takin' something that keeps me well. I class it the same as eating food and drinking water. I need that to keep me level and keep me… I feel totally normal now whereas I couldn't before and I was on all of the old medications […] whole range of different and they never really worked. Clozapine was the first one that ever worked. The only downside to it was the side effects; the dribble, makes you sleep, makes you weight gain, things like that but when you weigh up the good and the bad Clozapine is up there it does the job." Adam, CSU

### Playing my part
Forensic HCPs appreciated the therapeutic value of supporting CSUs to be as independent as possible through normalisation and socialisation. They perceived that a large part of their role was to work with and support CSUs in this way, rather than focussing solely on the therapeutic use of medication.

"A large part of our job isn't just about giving medication to people. (What) I quite like about my job is about socialisation and normalisation and trying to get people to live in a normal way." John, Forensic CPN

In contrast, General Adult HCPs perceived a sense of responsibility for coordinating the process of clozapine supply and undertook as much of the process as possible on behalf of the CSU. They only saw shared-care clozapine as an alternative method of obtaining a supply of clozapine.

"Difficult I would say, very difficult, lots of chasing finding about results, seeing if the pharmacy understands you and knows what you're trying to ask for, and then trying to establish whether they've actually got the medication or not, when they're going to get it then you've got to faff around, pick it up and get it to the patient it's a nightmare." Kate, General Adult CPN

GPs perceived their main role in shared care to be prescribing clozapine, checking FBC results and CSU well-being and functioning. However, they also spent time liaising with primary and secondary care team members with respect to any CSU or clozapine-related problems.

"…so there's the practicalities of just the physical monitoring of whether the bloods are ok umm there is a responsibility just to see the patient is ongoing wellbeing and mental health is stable and they're functioning. The responsibility to support the people taking the blood and if they're concerned, they're obviously not qualified, don't have the expertise to make any sort of mental health type assessment, so being available for them to be able to come and say this isn't right worried about this and then obviously responsibility for liaising with the mental health team if there are concerns or worries." Dr White, GP

In order to supply clozapine on prescription, community pharmacists require all previous steps of the process to be completed. In practice, community pharmacists often found they were heavily involved in identifying and coordinating missing elements of the process. which prevented them from completing their dispensing role.

"So we do all the chasing about to find out what stage we're at with all these things because if you phone Denzapine they'll say 'oh we haven't had a result in yet' and then you have to phone the surgery say, 'has this person had a blood test' or get in touch with CPN so we're doing a lot of chasing around." Charles, community pharmacist

CSUs perceived their role in shared care was to obtain a FBC from the GP surgery, order their prescription and collect and take their clozapine. Many CSUs used services provided by community pharmacy such as repeat prescriptions and delivery to support them in undertaking their role in shared care.

"Yeah it all good for me now on, so all I've got to do is every 28 days have my bloods done and then they (community pharmacy) do the rest and that, so I don't have to sign no paper or run around no more I've just got to go to the GP surgery and see the nurse and have blood samples." Simon, CSU

## Multi-professional Relationships
Being recognised as a member of a particular team supported the development of multi-professional relationships and communication. These relationships were integral to building trust and understanding of each other's role and the provision of peer support.

### Recognition as a team member
Knowing the members of the team involved in the provision of shared-care clozapine positively influenced communication and supported building relationships with both CSUs and HCPs being more comfortable to

make contact with someone if they knew them. This increasing familiarity led to increased involvement and contact, which developed relationships still further.

"I think if there was a problem with one of my guys in the (community) pharmacy where I've been in I'll actually go in and talk to him (community pharmacist) now whereas before I'd come through (the hospital pharmacy) so I talk to him now." John, Forensic CPN

Through building a relationship, individuals were increasingly able to understand the expertise and role each team member held, which supported identifying the most appropriate person to contact for support.

"I mostly do things to do with my mental health with my consultant and things to do with my physical health with my GP." Richard, CSU

### Supporting each other
Relationships were a key element in feeling supported and building trust. Support ranged from provision of information or advice to provision of additional resource when needed. The sense of feeling supported was characterised through a reliable relationship which was responsive to need.

"Knowing that there is somewhere that you can call on is important and certainly the particular community mental health nurse who is involved in the care she is excellent. Apart from these meetings she is available for discussion and she has come along and organised blood tests and like if there seems to be a more urgent situation where some compliance has gone wrong." Dr White, GP

Sharing the responsibility of clozapine provision through shared care was seen as a source of support for primary care by GPs and RCs, as it reinforced a sense of an equally shared responsibility and supported the ability to work in a multi-disciplinary way.

"Well it enables us to umm to manage this difficult client group for patients that need to take it, so we are happy to accept our share of the responsibility for that and… so it is a positive thing and it helps us with our sort of multi-disciplinary work." Dr Hudson, GP

## DISCUSSION AND CONCLUSIONS
### Principle findings
All participants perceived clozapine provision as different to other antipsychotics because of the process of supply, its beneficial and negative effects, greater reliance on members of the multi-professional team and system processes. Greater knowledge and understanding of clozapine provision was generally obtained through experience and led to individuals having more confidence in their own role and resolving issues that arose. Developing multi-professional relationships was key in building

trust, understanding roles, improving communication and feeling supported. Shared care reinforced a sense of equal and shared responsibility between primary and secondary care; this supported further development of multi-professional relationships and was seen as a source of support. Addition of a GP to the team provided more holistic care for the CSU, enabling physical health interventions and improved GP awareness that their patients were taking clozapine. This awareness decreased potential iatrogenic health risks associated with clozapine.[13]

Surrogate markers of stability and capability were originally used to identify CSUs who were potentially suitable for clozapine shared care. However with experience, Forensic HCPs identified shared care as a new opportunity to demonstrate trust in the CSU and support them to develop skills of self-dependence and independence through normalisation and socialisation. This reduced Forensic HCPs reliance on surrogate markers. Engaging in shared-care clozapine led to a change in the relationship dynamic between the Forensic HCPs and CSUs, which in turn enabled the CSU to take responsibility and ownership for their care. In contrast, General Adult HCPs held paternalistic assumptions of CSU abilities and/or expectations in relation to their independence and involvement in managing their own medication. This affected General Adult HCPs ability to trust CSUs in possibly taking on this responsibility or perceiving a need for shared-care clozapine.

## Implications for practice
### Relationships in healthcare provision
In 2010, the Health Foundation launched 'Closing the Gap through Changing Relationships', a programme which focused on recognising the need to change the way healthcare systems work by challenging beliefs and behaviours of healthcare workers to improve quality of care. Evidence demonstrates that best health outcomes and experiences are achieved when people have an active part in their own care and receive responsive support according to their needs. Being able to play an active role is a consequence of the dynamic created by the manner in which support is provided.[14] The programme also explored relationships between service users and providers in healthcare settings using a conceptual framework proposed by Wish et al,[14 15] which identified four primary relationship dimensions to describe the character of different relationships. These are power, valence, intensity and formality.[15]

We mapped the HCP–CSU relationships in Forensics and General Adult to the above relationship dimensions to explore how shared-care clozapine influenced these HCP/CSU relationships. See table 4 for further details.

General Adult HCPs appeared to hold negative assumptions of the abilities and/or expectations of CSUs managing their own medication. These assumptions reinforced their inherent paternalistic role demonstrated by taking responsibility for the CSU obtaining clozapine, which created dependence. Shared-care clozapine

appears to have had no influence in supporting changes in the General Adult HCP–CSU relationship dimensions and so it continues to be characterised by asymmetric, competitive, distant and professional qualities. This may provide an explanation as to why General Adult HCPs did not appear to see a need for shared-care clozapine or invite CSUs to participate in this study.

In contrast, the effect of shared-care clozapine on Forensic HCP–CSU relationship influenced change in all four Wish dimensions. Engaging in shared-care clozapine enabled a shift in the power differential between HCP and CSU to a more symmetric and cooperative approach, possibly through the ability to trust in each other and share a common agenda of CSU independence. The willingness to engage in new ways of working enabled them to tailor their support for CSUs to continue to develop skills for independence, demonstrating commitment to the long-term agenda of independence and relationship integrity. Shared-care clozapine appeared to cultivate changes in the Forensic HCP–CSU relationship dimensions, which enabled CSUs to expand their ability to live independent and fulfilling lives; key components of person-centred care.[16] In contrast, the General Adult HCP–CSU relationship is reflective of a biomedical l model and they described a more negative experience of shared care compared with the more positive experience of Forensic HCPs.[17]

### Person-centred care
Providing care which is person centred is dependent on ensuring care affords people dignity, respect and compassion; care, support or treatment which is coordinated, personalised and enables patients to live an independent and fulfilling life.[16]

Key principles of person-centred care with particular relevance to successful shared-care clozapine outcomes include knowing the patient as a person, recognising their individuality and expertise in their own health and care and taking a holistic approach to assessment and provision of care which is centred around them. Importantly, staff need to be supportive, share the power and responsibility of care and be well trained in communication.[18]

Engagement in shared-care clozapine with a relationship demonstrated by the Forensic HCP/CSU partnership provides an example of a shift in care from paternalistic to person centred. We suggest that shared-care clozapine in this context offers an opportunity to demonstrate dignity, respect and compassion to CSUs through trust and handing over greater responsibility to the CSU. The provision of coordinated care and support through the development of relationships between multi-professional team members in primary and secondary care enabled personalised care and support for CSUs, which subsequently reflected in CSUs developing their ability to manage their own role in shared care. This person-centred approach ultimately enabled CSUs to live an independent and fulfilling life by self-managing their own clozapine.

**Table 4** Mapping relationships to the Wish model

| The Wish model relationship continuum components | | The relationship ynamic | |
|---|---|---|---|
| | | Forensic community mental health team (CMHT) healthcare professionals (HCPs) | General Adult CMHT HCPs |
| Symmetric – Asymmetric | *Power* dependence centredness agency | Greater sense of partnership for both clozapine service users (CSUs) and HCP. Themes centred around the CSU rather than the HCP. CSUs have gained greater agency and reduced dependence on HCPs. *Result: perception of a symmetrical HCP/ CSU relationship.* | HCPs complete activities either for or on behalf of CSU. Themes centred on HCPs rather than CSU. CSU appears to be highly dependent on the HCP. *Result: perception of an asymmetrical HCP/CSU relationship.* |
| Cooperative – Competitive | Valence agreement communication conflict | HCP and CSU agendas aligned to increase independence and normalisation. Engagement in shared care provided an opportunity to achieve their shared agendas. HCPs trust in CSUs ability to perform own role in shared care. *Result: perception of a cooperative HCP/ CSU relationship.* | No evidence of agenda alignment. Formal medical knowledge competing with lay knowledge. Some HCPs described lacking trust in CSU relationships. *Result: perception of a competitive component to HCP/CSU relationship.* |
| Intimate – Distant | *Intensity* transactional/ relational commitment | Monitoring prescription collection transactional element but in context of a relational focus on a CSU long-term goals. Forensic HCPs demonstrated commitment through motivation and willingness to adapt their beliefs and behaviours to fulfil the shared agenda of independence. CSUs committed to their own role in shared care. *Result: perception of an intimate HCP/CSU relationship.* | Adult HCPs talked about undertaking activities on behalf of the CSU with no reference to the future aims of the CSU. Adult HCPs demonstrated little change in beliefs and behaviours in a shared-care role of the CSUs, suggesting a distant relationship with little commitment for the potentially longer term goals of the CSU. *Result: perception of a transactional and distant HCP/CSU relationship.* |
| Social – Professional | *Formality* affect exclusivity | Engaging in shared care enabled a shift in formality dimension to social as the relationship extends into primary care. Social affect aided by the commitment to HCP–CSU relationship. *Result: perception of a more social professional HCP/CSU relationship.* | Considerable professional formality. Professional affect demonstrated by the power differential between the HCP and CSU. *Result: perception of a formal professional HCP/CSU relationship.* |

Dartmouth Atlas defines effective care as *'services that are of proven value and have no significant tradeoffs'*.[19] Based on CSUs and Forensic HCPs experiences, shared-care clozapine is effective for these participants. Currently, there is no definition of acceptability in healthcare, although Sekhon *et al* (2017) propose a theoretical yet to be validated framework.[20] We suggest that acceptability of shared-care clozapine is demonstrated by the active engagement of CSUs and Forensic HCPs in the process.

### Future research

Future research using implementation theory is needed to understand in greater detail the barriers and enablers to wider implementation of shared-care clozapine to support integration of people with serious mental illness into the community.[21 22]

### Strengths and limitations of this study

This is the first qualitative study to explore individual perceptions on experiences of shared-care clozapine service by those who use and provide it. Identification of potential participants relied on two information sources being up to date: the DMS website and electronic medical notes. CSU participants were not equally represented from Adult and Forensic CMHTs, which may mean experiences and perceptions of General Adult CSUs are not sufficiently represented. Recruitment of CSUs was dependent on invitation for participation from their own clinical team. Greater recruitment of Forensic CSUs could relate to their enabling approach to care or a reflection of the team's experience of shared care in comparison with General Adult CMHT. No carers of CSUs participated, possibly reflecting the isolation that some people with serious mental illness live with.

IPA methodology guided the development of topic guides, completion of interviews and homogenous focus groups used to collect the qualitative data. IPA also allowed the experience of the researchers to be noted in the reflexivity associated with data analysis. Quality assurance processes were completed to provide validity to this research, for example, piloting topic guides, supervised focus groups and joint co-coding and re-coding of data. Data validity is reflected in the ability to substantiate the analysis with participant quotes and provide a dialogue of discussion between the results and existing literature.

## CONCLUSIONS

Mental health has seen an increase in demand on limited health service resources with cuts in funding. This has led to a number of trusts embarking on transformative programmes in order to reduce costs, shift demand from acute services and deliver care focused on recovery and self-management.[23] Person-centred care is recognised nationally as one way in which this can be achieved through a substantial effect on quality of care.[16][18] Our results suggest shared-care clozapine could be one way of expanding person-centred care in mental health which the literature demonstrates demonstrable improvements in quality of care, integration into the community and reductions in stigma and healthcare resources.

**Acknowledgements** We thank all study participants for their input into this study and for the team members who helped in identification of potential participants.

**Contributors** CS and DT—substantial contributions to the conception, design, acquisition, analysis and interpretation of data for the work; drafting and critically revising the work; final approval of the version to be published and agreement to be accountable for all aspects of the work in ensuring that questions related to the accuracy or integrity of any part of the work are appropriately investigated and resolved.

**Funding** This study was funded by a small grant from the College of Mental Health Pharmacy.

**Competing interests** Both authors have completed the ICMJE uniform disclosure form at www.icmje.org/coi_disclosure.pdf and declare: CS reports grants from College of Mental Health Pharmacy (CMHP), during the conduct of the study. DT reports that at the time of the grant being awarded was a council member of the CMHP, but the award allocation was judged by a grant advisory panel with due process and also completed two advisory boards for Sunovion (Lurasidone) during this time; and is a trustee for Bath Mind.

**Provenance and peer review** Not commissioned; externally peer reviewed.

**Data sharing statement** All data collected during this study have been included in the analysis and published.

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
