## [Reviewer comments · BMJ Open]

ARTICLE DETAILS

TITLE (PROVISIONAL)	Cross Sector User and Provider Perceptions on Experiences of Shared Care Clozapine – a qualitative study
AUTHORS	Sowerby, Camilla; Taylor, Denise

VERSION 1 - REVIEW

REVIEWER	Karin Sjögren Department of Nursing, Umeå University, Sweden
REVIEW RETURNED	16-May-2017

GENERAL COMMENTS	This is an interesting study but the paper needs some major revisions before publishing the paper: I have some concerns about the aim, the method and the result. The aim is to explore experiences and perspectives, and to gain an understanding of the effectiveness and acceptability of the shared-care service. Yet the method of analysis used is interpretative phenomenological analysis where the focus is to elicit meanings from participants' narratives. In the result there are mostly descriptions of experiences, not so much about meaning. The wording of the subordinate themes seems more like categories of experiences than meanings: clozapine process, sharing of care, clozapine process, provision of care, multi-professional relationships. Is interpretative phenomenological analysis an appropriate method of analysis in this study and does the result elicit meanings from participants' narratives? This needs to be discussed under strengths and limitations. Abstract There are different aims/objectives in the abstract and in the text. In the abstract there are only one aim, but in the text there are two aims. This has to be corrected. Result: It is hard to understand the logic of the superordinate themes, and what is described under the themes. It is also hard to see the meaning from the participants' narratives in the wording of the superordinate themes. For example: clozapine process: Includes importance of knowledge of the process, learning by experience is important, how HCP decide whether a CSU would be suitable for shared-care, how the shared care model facilitate the true potential of the patient. Shared of care: Includes shifted responsibility, the need to work in a new way, CSU experience independence and boosted self-esteem, change in relationship, CSU in the center of care, assumptions limited the HCP's trust in CSU ability to take responsibility, stability
--

	and capability are used when to decide if a CSU would be suitable for shared-care... The provision of care: Includes different roles, attitudes and responsibility of each member of the MDT, different ways of finding time to understand shared-care, and CSUs roles in the shared-care. It also includes CSU and professionals thoughts about the clozapine-treatment compared to other treatments. My recommendation is to develop the result further and to use other sub-themes. All to make it easier to understand the meanings of the experiences and perspectives that has been narrated. Now you have to read the result many times and make your own analysis in order to understand. The text in the principle findings in the discussion could be a starting point for how to organize the result section. Result and discussion The second aim was to gain an understanding of the effectiveness and acceptability of the shared-care service. Little in the result or in the discussion describes acceptability and effectiveness. As the understanding of the effectiveness and acceptability of the model is one of the objectives of the study this needs to be developed further. Discussion In the abstract it is written: the Wish conceptual model of relationships provided further insight into the acceptability of care delivery. But in the discussion where the model is referred to, I find no comment about acceptability. This has to be developed further. In discussion: page 9 line 4, the Forensic HCP/CSU relationship appears to possess a more symmetric co-operative, intimate and social characteristics if mapped onto the Wish conceptual model of relationship. Please develop this further how the result show a more symmetric co-operativerelationship between HCP and CSU! In discussion: page 9 The characterization of the Forensic HCP/CSU could be said to fulfil all of the conditions of the enabling principle of person-centered care.....Please develop this further what the principles are and what in the result that fulfill the conditions. The title of the paper is: Enabling person-centred care through shared-care clozapine so this needs some more attention in the text.
--	--

REVIEWER	Inti Qurashi Hon senior lecturer consultant forensic psychiatrist Institute Brain and Mind University of Manchester
REVIEW RETURNED	19-May-2017

GENERAL COMMENTS	The study aims i) to explore perceptions and experiences of people receiving and or delivering a shared care clozapine service and ii) gain an understanding of the effectiveness and acceptability of a shared care clozapine service. The study appears compromised in its' aims by a) The selection of participants; of the 38 identified 32 were Healthcare professionals (I could find no breakdown by profession in terms of numbers or service in which they worked (general adult or forensic) which would have been informative. I am unsure why a receptionist was included as a participant b) Only 6 clozapine service users were identified of which 5 were
--

	forensic patients; given the authors principle findings regarding a general adult/forensic service difference this may be explained by the bias of participants. This needs to be reflected in the discussion and limitations. c)The principle findings section requires major revision to reflect the actual findings - an interesting point that appears to be overlooked is the observation that within general adult services patients on shared clozapine care plans cannot be discharged and it potentially prevents patient independence; this raises the spectre of unintentional consequences in terms of increasing caseload and work for general adult CPN's and whether shared care clozapine is deliverable within devolved commissioning budgets and not least shared care clozapine contributing to iatrogenic harm in terms of limiting independence and progression of patients. d) I apologise if I missed it; I could find no definition of effectiveness nor a measure of it. e) Given the importance of GP's in shared care clozapine I am surprised none were interviewed; their views would have been I thought necessary. f)The potential difficulties of shared care clozapine are not considered in terms of drug errors; I would have expected to see this in this paper e.g 'Clozapine and concomitant medications: Assessing the completeness and accuracy of medication records for people prescribed clozapine under shared care arrangements.'Murphy K, Coombes I, Moudgil V, Patterson S, Wheeler A. Eval Clin Pract. 2017 May 4. doi: 10.1111/jep.12743. g) to a reader unfamiliar with the field/topic I found the paper difficult to read in terms of complex phraseology being used when simple sentences would have been clearer.
--	--

VERSION 1 – AUTHOR RESPONSE

The objective of this study was to explore individual perceptions on experiences of people receiving and/or delivering a shared-care clozapine service. Interpretative phenomenological analysis methodology was used to explore individual experiences of clozapine service users (CSU), general practitioners (GPs); community psychiatric nurses (CPNs), social workers (SWs), community and hospital pharmacy staff and responsible clinicians (RCs). Data was collected using semi-structured interviews and focus homogenous groups. This study highlights differences between Adult and Forensic healthcare professionals (HCPs) engagement in shared care clozapine.

Both Adult and Forensic HCP-CSU relationships were mapped to the Wish conceptual framework of relationships to provide insight into how shared care clozapine can provide a mechanism for provision of person-centred care. Person-centred care was present in the Forensic HCP-CSU but not General Adult HCP-CSU relationship. We propose that wider implementation of shared care clozapine could enable greater integration of people with serious mental illness into the community, reducing stigma, whilst improving patient outcomes associated with person-centred care.

This manuscript should be published in the BMJ Open as the implications for practice include: the need to reflect on the impact of service provision on CSUs ability to recover and rehabilitate; and how shared care clozapine could provide a mechanism for provision of person-centred care, leading to

improvement in quality of care and potentially decreasing demand on limited health service resources. We have responded to each of the reviewer comments as detailed in the Excel Spreadsheet embedded below and have taken the opportunity as suggested, to re-write the results section with greater clarity and focus.

This manuscript is over the suggested word count due to the need to publish quotes as evidence of data synthesis, which cannot be tabulated without disrupting the flow of the reader. This has been discussed with yourself by email (attached as a supplementary file) demonstrating agreement that the word count is acceptable.

VERSION 2 – REVIEW

REVIEWER	Karin Sjögren Department of Nursing, Umeå University, Sweden
REVIEW RETURNED	28-Jul-2017

GENERAL COMMENTS	The revisions have resulted in improvements from the previous submission, and I only have some concern about the aim/s. In the abstract there is one aim: To explore individual perceptions on experiences of people receiving and/or delivering a shared-care clozapine service. In the background there are two aims: 1) To explore individual perceptions on experiences of people receiving and/or delivering a shared-care clozapine service. 2) To gain an understanding of effectiveness and acceptability of shared-care clozapine. Both aims should be included in the abstract.
---

REVIEWER	inti qurashi University of Manchester UK
REVIEW RETURNED	21-Jul-2017

GENERAL COMMENTS	The substantial revisions have improved the quality of the paper making it more focussed and readable.
--

VERSION 2 – AUTHOR RESPONSE

Please find uploaded a revision of the manuscript which addresses the comments made by a reviewer to ensure that the abstract reflects both objectives stated in the background.